# Long Non-Coding RNAs Expressed in the Peanut Allergy for Understanding the Pathophysiology of Peanut Allergy Rat Model

**DOI:** 10.3390/foods11233760

**Published:** 2022-11-22

**Authors:** Manman Liu, Sen Li, Boya Li, Shanfeng Sun, Guirong Liu, Junjuan Wang, Mengzhen Hao, Huilian Che

**Affiliations:** 1Key Laboratory of Precision Nutrition and Food Quality, Key Laboratory of Functional Dairy, Ministry of Education, College of Food Science and Nutritional Engineering, China Agricultural University, Beijing 100083, China; 2College of Agronomy and Biotechnology, China Agricultural University, Beijing 100193, China

**Keywords:** peanut allergy, long non-coding RNAs, logistic regression, diagnostic model

## Abstract

Background: Peanut allergy (PA) has become a clinical and public health problem, which is mainly regulated by genetics, immune responses, and environmental factors. Diagnosis and treatment for PA have always remained huge challenges due to its multiple triggers. Studies have shown that long non-coding RNAs (lncRNAs) play a critical role in the development of allergic diseases. Method and Results: In the current study, we examined the plasma lncRNA expression profiles of peanut allergy Brown Norway rats and healthy controls and 496 differently expressed lncRNAs were identified, including 411 up-regulated genes and 85 down-regulated genes. We screened 8 lncRNAs based on the candidate principle and the candidates were verified in individual samples by quantitative real-time PCR. Then, the four lncRNA-based diagnostic model was established by least absolute shrinkage and selection operator (LASSO) and logistic regression, which was proved by area under the receiver operating characteristic curve (AUC). Conclusions: In summary, we assessed the correlation between lncRNA expression levels and the diagnosis of peanut allergy, which may perform a vital role in guiding the management of peanut allergy.

## 1. Introduction

As one of the major public health problems worldwide, food allergy (FA) is thought to be the second-wave allergy epidemic after asthma, with evidence of increasing prevalence [1]. It is reported that food allergy affects 2% of adults and 8% of school-age children [2]. There have been many explorations in food immunotherapy for food allergy over recent years but side effects and safety of the treatment have remained a concern [3,4]. The management of FA mainly depends on strictly controlling the allergens that may cause adverse reactions in contact with our body, including the big eight (peanuts, nuts, eggs, soybeans, fish, shellfish, milk, and wheat) [5]. Peanut allergy (PA), considered one of the most serious food allergies, generally does not disappear with age. Only upon exposure to a small number of peanut allergens do adverse immune responses arise, causing acute symptoms, such as hives, pruritus, respiratory difficulty, cardiovascular compromise, and gastrointestinal disturbance, some of which can be life-threatening [6,7]. Developing therapies for peanut allergy continues to be a challenge and our understanding of its etiology and pathobiology remain limited.

Anaphylaxis is generally believed to be affected by a combination of genetics, immune responses, and environmental factors [8]. Some gene expression studies in allergic individuals contribute some ideas to the potential molecular mechanisms [9,10]. For example, altered expression of genes involved in helper T (Th) cell cytokine signaling were revealed in circulating peripheral blood mononuclear cells of fruit- and/or latex-allergic subjects [9]. So, in the past, the dysregulation of protein-coding genes was believed to work for the occurrence and development of food allergy [11] but recently accumulated data have reported the potential role of ncRNAs in almost all physiological and pathological activities of humans [12].

Long non-coding RNAs (lncRNAs) were defined as non-protein-coding RNA transcripts with greater than 200 nucleotides in length, accounting for approximately 98% of the total RNAs. Participating in considerable biological processes though chromatin remodeling, cell cycle regulation, splicing regulation, mRNA degradation, and others, lncRNAs are reported to contribute to the emergence and development of dozens of diseases [13,14,15]. The role of some non-coding RNAs in allergic diseases, including asthma and atopic dermatitis, has been demonstrated, indicating that they can be served as therapeutic targets and biomarkers for allergic diseases [16]. However, little emphasis has been placed on characterizing aberrant lncRNAs that mediate peanut allergy.

To date, peanut allergy is clinically confirmed by in vivo methods, such as food challenges and skin prick tests (SPT), combined with serum-specific immunoglobulin E (sIgE) detection, which tends to overdiagnose for PA [17]. The developments of multi-omics technologies make proposing novel biomarkers for the diagnosis of PA possible. Therefore, in the present study, to identify differentially expressed lncRNA patterns and elucidate their potential roles in peanut allergy, we constructed a peanut allergy rat model and performed a systematic transcriptome analysis derived from deep transcription sequencing between PA individuals and a control. Moreover, on the basis of aberrantly expressed lncRNAs, a lncRNA-based signature was established for predicting the peanut allergy prognosis and understanding the pathophysiology of peanut allergy. Moreover, the lncRNA–mRNA co-expression network was investigated to find the underlying mechanisms, which may provide new insights into exploring the food allergy-induced inflammatory response.

## 2. Materials and Methods

### 2.1. Allergic Rat Models

In total, 53 five-week-old female Brown Norway rats were purchased from Vital River Laboratories, Inc. (Beijing, China) and raised in a specific pathogen-free (SPF) animal laboratory at a temperature of 23 ± 1 °C, humidity of 55 ± 5%, a 12 h light/dark cycle, and air exchanges at 15 times/h. Free food and water intake and adaptive feeding was allowed for a week prior to initiation.

Our study consisted of two stages. In the training set, 41 rats were divided into two groups with equal body weight and the numbers of control (Con) and peanut allergy (PA) groups were 18 and 23, respectively. As shown in Figure 1A, the PA group rats were subjected to intraperitoneal injection (i.p.) of 0.5 mL 4 mg/mL peanut protein solution with 125 µL Alum Adjuvant on Days 1, 3, 5, and 15. Then, the sensitized rats were challenged intraperitoneally with a 5-fold dose of peanut protein on Day 21. Phosphate-buffered saline (PBS) was given to the control group by i.p. at an equal capacity. The validation set falls into four groups of control, peanut allergy, shrimp allergy, and milk allergy (N = 3), in which the rats were injected intraperitoneally with different kinds of allergen protein to construct food allergy rat models, as mentioned above. 

All animal experiments complied with the relevant guidelines and was approved by the China Agricultural University Animal Experimental Welfare and Ethical Inspection Committee.

### 2.2. ELISA

The blood samples, collected from the sacrificed rats 45 min after the allergen challenge, were centrifuged at 3500 rpm for 15 min after standing for 2 h to decant the serum and then were stored at −20 °C. Commercial rat ELISA kits (Abcam, Cambridge, MA, USA) were used to quantify the relative levels of specific IgE, IgG_1_, histamine, MMCP-1, IL-4, and IFN-g from obtained serum samples.

### 2.3. RNA Extraction and Transcription Sequencing

Total RNA extraction from whole blood samples was conducted using the TRNzol reagent (TransGen Biotech, Beijing, China). Then, the RNA concentration was determined by a NanoDrop One spectrophotometer (Thermo Scientific, Waltham, MA, USA) and RNA quality was evaluated by an Agilent 2100 Bioanalyzer (Agilent Technologies, Palo Alto, CA, USA). After the performance of quality control, at least 3 μg of RNA with an RNA integrity number (RIN) greater than 8.0, and with A260/A280 located in the middle of 1.8–2.2, was used for database construction.

Complementary DNA (cDNA) libraries were prepared using a Next Ultra RNA Library Prep Kit (NEB, Franklin, NJ, USA) and total RNA with end repair and adapter ligation. A 100 bp paired-end RNA sequencing experiment was performed using the Illumina platform. Clean reads were obtained by removing the adaptor for low-quality raw reads for quality control. Using HISAT2 (v2.1.0) [18] with default parameters, paired-end clean reads were mapped and StringTie (v1.3.3b) was adopted to assemble transcripts. LncRNAs and mRNAs were annotated with the given database and calculated by fragments per kilo-base per million reads (FPKM) for quantitative analysis. The DESeq2 package was used to identify the differentially expressed genes (DEGs) with *p* < 0.05 and |log2(fold change)| > 1. The database construction and sequencing services were provided by Metware (Wuhan, China).

### 2.4. qRT-PCR

Total RNA separated from whole blood samples was reverse-transcribed using a cDNA reverse transcription kit (Vazyme, Nanjing, China) according to the manufacturer’s instructions. To validate the expressions of candidate lncRNAs, quantitative real-time PCR (qRT-PCR) analyses were performed by TransStart^®^ Green qPCR SuperMix (Bio-Rad Laboratories, Hercules, CA, USA). The relative lncRNA expression level was normalized to the β actin expression and calculated using $2^{-(Ct_{target}-Ct_{reference)}}$. The primer pairs used for the amplification of the target genes are presented in Appendix A.

### 2.5. Bioinformatics Analysis

Construction and validation of the four-lncRNA signature: The candidate lncRNAs of the training set determined by transcription sequencing were exposed to least absolute shrinkage and selection operator (LASSO) analysis for filtering out the prognostic lncRNA biomarkers. Then, the screened variables were identified to construct a risk signature for peanut allergy using multiple logistic regression (MLR) by the “rms” package [19], whose coefficients indicated their impact on PA prediction. The R package “ROCR” was used to generate the receiver operating characteristic (ROC) curve [20] and the area under the curve (AUC) of the ROC was compared to inspect the performance of the constructed diagnostic model based on lncRNAs.

Prediction and functional enrichment analysis of lncRNA target genes: It is reported that lncRNA functions by regulating mRNAs [21]. lncRNA target genes were predicted by their positional relationship (cis) and expression correlation (trans) with protein-coding genes [22]. We set the threshold for positional relationship-based target gene screening at 100 kb upstream and downstream of the lncRNA. To better understand the potential function of the predicted target genes and differently expressed genes, we used the “clusterprofiler” package of R to perform gene ontology (GO) function enrichment analysis and Kyoto Encyclopedia of Genes and Genomes (KEGG) pathway enrichment analysis. Statistical significance was set at *p* < 0.05.

Analysis of lncRNA–mRNA co-expression networks: The co-expression networks of selected lncRNAs and DEGs were conducted by Pearson’s correlation coefficient (PCC) using the “corrplot” R package based on the normalized RNA-seq signal intensity. Significant correlation pairs with *p* < 0.01 and PCC > 0.95 were chosen to construct the network.

### 2.6. Statistical Analysis

All data in the experiment were analyzed using SPSS 14.0 and GraphPad Prism 8 software. The data were analyzed using two-tailed tests and expressed as mean ± SD from independent biological replicates. *p*-value < 0.05 was considered statistically significant.

## 3. Results

### 3.1. lncRNA Expression Profiles in the Blood of Peanut Allergy Rats

To determine whether peanut allergens influence the expression of lncRNAs in vivo, the peanut-induced allergic rat model was established and the peanut allergy symptoms were evaluated after the challenge on Day 21 (Figure 1B–E). A significant decrease in body temperature occurred in the peanut allergy group rather than the control group. Moreover, the peanut-specific IgE and IgG_1_ levels of PA rats significantly increased compared to the control group. As previously reported, the peanut allergen treatment caused IL-4 elevation and an IFN-γ decrease [23]. Additionally, above all, the levels of histamine and MMCP-1 measured using ELISA significantly increased in the peanut allergy group compared to the control group, which demonstrates a successful establishment of a peanut allergy rat model.

To explore the potential functions of lncRNAs involved in the progression of peanut allergy, we performed a deep transcriptome sequencing in the training set, including three control and three peanut allergy samples. A total of 6342 lncRNAs were identified based on the length and capacity of coding protein predicted by CNCI, CPC2, and PLEK, with 35.5% lincRNA, 12.5% antisense lncRNA, and 52.0% intronic lncRNA involved (Figure 2A). Hierarchical clustering showed a significant difference in the blood levels of lncRNAs. The differentially expressed lncRNAs (DELs) were used to generate a heatmap and volcano plot, as shown in Figure 2B,C. Based on the criteria of *p* < 0.05 and log2 (fold change) > 1, we found 496 aberrantly expressed lncRNAs between the two groups, including 411 up-regulated and 85 down-regulated lncRNAs. Appendix A depict the top 20 up-regulated and down-regulated lncRNAs. 

Aimed at exploring the functions associated with these DELs, target genes were predicted from cis- and trans-regulation, which were then used to perform GO and KEGG enrichment analyses as depicted in Figure 2D,E. GO analysis mainly concentrated on the transcription coactivator activity, mitochondrial membrane parts, antigen processing and presentation of peptide or polysaccharide antigens via MHC class II, and MHC class II protein complex, among others. KEGG analysis indicated endocytosis, amyotrophic lateral sclerosis, viral carcinogenesis, and other pathways.

### 3.2. Construction and Validation of the Four-lncRNA Signature

To build more precise diagnostic models, candidate lncRNAs were identified from the deep sequencing (N _control_ = 3, N_PA_ = 3) to meet the following criteria: (1) When compared to the control group, PA individuals had up-regulated expression of the candidate lncRNAs with a log2 (fold change) > 1 and *p* < 0.05; (2) lncRNAs were annotated from known gene databases; and (3) the average relative expression of lncRNAs was validated by qRT-PCR > 0.01. In addition, eight candidate lncRNAs met the criteria: ENSRNOT00000093217 (X3217), ENSRNOT00000087227 (X7227), ENSRNOT00000057756 (X7756), ENSRNOT00000090335 (X0335), ENSRNOT00000085965 (X5965), ENSRNOT00000081904 (X1904), ENSRNOT00000074532 (X4532), and ENSRNOT00000075924 (X5924) (see Table 1). These eight lncRNAs were verified in individual samples in the training phase by qRT-PCR (Figure 3A). Of the eight lncRNAs examined, X5965, X1904, X4532, and X5924 showed aberrantly increased expression in the blood of peanut allergy rats compared to ordinary rats.

Then, these candidate lncRNAs were screened for LASSO analysis, which indicated a set of four lncRNAs that seemed appropriate as biomarkers (Figure 3B). So, based on the qRT-PCR data from the training set, a discriminant model of an lncRNA panel was constructed to predict the diagnostic probability through logistic regression, which is shown in Table 2. According to the lncRNA-based prognostic score involved in the combined model, a nomogram was built (Figure 3C). Moreover, the calibration curve in Figure 3D also shows our model performing well in predicting whether an individual has a peanut allergy. To further individually evaluate the potential value of the candidate lncRNAs or to evaluate the constructed model as a peanut biomarker, ROC curve analyses was performed on the training set and test set, respectively. As can be seen in Figure 3E, the AUC values of X7227, X0335, X5965, X1904, and X4532 were all above 0.8, which shows a favorable predictive power. Meanwhile, the AUC value was 0.9433 when the panel of four lncRNAs was evaluated, which was better than any single lncRNA.

Finally, in the test stage, milk allergy (MA) and shrimp allergy (SA) rat models were additionally established. The ROC curve was used to determine the potential value of our logistic regression equation to distinguish subjects with and without PA. The AUC values are shown in Table 3. The model of the panel of the four lncRNAs generated an AUC of 0.75, 0.667, and 0.75 in the PA and control groups, PA and MA, and PA and SA, respectively, which indicates a positive effect on predicting the probability of diagnosis with PA.

### 3.3. Building of mRNA Expression Profile 

In order to investigate the potential relationship of transcripts co-regulating the peanut allergy process, the annotated mRNAs differentially expressed between the control and peanut allergy groups were screened and subjected to functional enrichment analysis. A total of 2123 differentially expressed genes were identified at a cutoff of |log2(fold change)| > 1, *p* < 0.05, including 1480 up-regulated mRNAs and 643 down-regulated mRNAs (Figure 4A,B). KEGG pathway analysis mainly concentrated on the pathways of neurodegeneration-multiple diseases, amyotrophic lateral sclerosis, Epstein–Barr virus 1 infection, and antigen processing and presentation (Figure 4C). Then, GO analysis was performed to explore the biological processes, cellular components, and molecular function, and identified the top 50 pathways, which were “autophagy”, “mitochondrion organization”, “chromatin”, “ubiquitin-like protein transferase activity”, and “antigen processing and presentation of peptide or polysaccharide antigen via MHC class II”, etc. (Figure 4D). 

### 3.4. Predicted Function of the Studied lncRNAs

It has been reported that lncRNAs may function as regulators of mRNAs, especially for nearby protein-coding genes [21]. Finally, for the sake of inquiring about the biological functions associated with the four lncRNAs selected for the diagnostic signature, we predicted the target genes of lncRNAs. In cis-regulation, we set the threshold for positional relationship-based target gene screening at 100 kb upstream and downstream of the lncRNA. A total of 22 cis target genes identified for X7227, X0335, and X1904 are exhibited in Figure 5A. The prediction of target genes trans-regulated by lncRNAs was implemented by correlation analysis of lncRNA and mRNA expression between samples. By using the Pearson’s correlation coefficient (PCC) method, with a threshold of a PCC absolute value greater than 0.95 and *p* < 0.01, we obtained 494 genes associated with X5965.

To gain insight into their underlying biological processes, we established lncRNA–mRNA co-expression network analysis in view of the data from lncRNA sequencing. The network analysis, presented in Figure 5B, was performed between the 4 lncRNAs and top 11 DEGs. The lncRNA–mRNA co-expression network revealed that X0335, X1904, and X7227 were highly correlated with *Hmces*, *Serpinb10,* and *Tmem256*, respectively, while X5965 coordinated with multiple genes, including *Micall2*, *Asns,* and others, indicating that the contribution of the 4 lncRNAs to peanut allergy may be through regulating the 11 DEGs.

## 4. Discussion

Peanut allergy (PA) is defined as an IgE-mediated immune response that usually manifests in childhood and leads to adverse health effects [24]. So, an accurate diagnosis can protects patients from future allergic and even severe life-threatening events. Clinically, diagnostic methods usually include a detailed clinical history, detection of peanut-specific sensitization by SPT and/or in vitro measurement of peanut-specific IgE, and a double-blind, placebo-controlled food challenge as the gold standard [25]. However, diagnosing PA is still a clinical conundrum given the frequent discrepancy between practical detection and clinical symptoms. Attempts have been made to create mathematical or computer models incorporating a range of factors. Klemens updated a model by adding allergic rhinitis, atopic dermatitis, and sIgE to peanut components Ara h 1, 2, 3, and 8 as candidate predictors, which showed favorable discrimination (88%) but poor calibration (*p* < 0.001) [26]. With emerging omics studies, the inhibitory transmembrane protein, a member of the immunoglobulin superfamily CD200R, has been identified as a highly up-regulated marker on peanut-specific T cells [27]. MicroRNA 193-5p, involved in the post-transcriptional regulation of IL-4, significantly down-regulated the PBMCs of milk-allergic individuals compared to nonallergic individuals [28]. However, a single variable tends to cause overdiagnosis. So, a diagnostic model consisting of multiple indicators is in urgent need to comprehensively predict the presence of peanut allergy.

In this study, we investigated the expression profile of blood lncRNAs in peanut-allergic rats, and constructed a model of four lncRNAs with a high potency to discriminate rats with peanut allergy from healthy rats. In the training stage, eight candidate lncRNAs captured from a smaller number of subjects were screened for testing on an expanded swarm by qRT-PCR. In our study, there was no difference in the expression of PVT1 [29], MAF [30], and GAS5 [31,32] (which have been reported to play a critical role in an allergic response), possibly due to our limited subjects or their engagement in the pathomechanism of peanut allergy in a tissue-specific manner. Emerging roles of lncRNAs came into the focus of research on hypersensitivity disease, for instance, asthma [33,34], allergic rhinitis (AR) [35], and atopic dermatitis (AD) [36], which were proven to be therapeutic targets and biomarkers of allergic disorders, while little is known about lncRNAs in the peanut allergy response. So, the mRNA expression profile was determined and bioinformatics analysis was performed in our study, which identified 2123 abnormally expressed genes. Among these, the gene for the low-affinity receptor for IgE (*FCER2*) has been implicated as a candidate for IgE-mediated allergic diseases and bronchial hyper-reactivity [37]. The activation of *FCER2* resulted in the down-regulation of IgE-mediated immune responses [38], with a significant down-regulation occurring in our results. In addition, the aberrant expression of the *MALT1* gene, implicated as an independent risk factor for peanut allergy [39], added authenticity to our study. 

lncRNAs have been demonstrated to serve as biomarkers for the diagnosis of cancer and cardiovascular diseases, among others [40,41,42]. Therefore, to evaluate the value of DELs as biomarkers for PA, we constructed a diagnosis model with a combination of four lncRNAs (X7227, X0335, X5965, and X1904) by logistic regression. According to the expression of lncRNAs between the PA and control groups in the training set, our signature based on lncRNAs demonstrated a higher AUC value (0.9433) and better specificity than when they were individually used. Additionally, what pleasantly surprised us was that the panel of the four lncRNAs could distinguish peanut allergy from milk allergy and shrimp allergy, the prevalence of which were more common [43]. 

However, limited research on the mechanisms of these lncRNAs in the process of peanut allergy has been undertaken. So, to gain a better understanding of the physiological role of the selected lncRNAs in PA, we explored the potential targets and pertinent pathways using bioinformatics. GO enrichment analysis demonstrated that antigen processing and presentation, MHC protein complex, cellular response to inorganic substances, and other pathways were associated with peanut allergy. lncRNA–mRNA co-expression network analysis showed a high correlation between the four lncRNAs and eleven mRNAs, including *Hmces*, *Serpinb10*, *Tmem256*, *Asns,* and others. A genome-wide association study on food allergy diagnosed by an oral food challenge first identified the serine protease inhibitor clade B (*SERPINB*) gene cluster as a susceptibility locus for food allergy, which is involved in immunological regulation or epithelial barrier function, emphasizing the role in food allergy [44]. 5-hydroxymethylcytosine binding, embryonic stem cell-specific protein (*HMCES*) (originally considered as a protein capable of binding 5-hydroxymethylcytosine (*5hmC*), an epigenetic modification generated by TET proteins) was strongly induced in activated B cells [45]. It was reported that HMCES deficiency leads to a significant defect in the class switch recombination of B cells [46], which contributes to the regulation of allergen-specific immune responses. Asparagine synthetase (*Asns*), considered a catalyst transforming the fate of Gln to asparagine, was a predictive biomarker in several human cancer types [47,48]. One study suggested an important role of *Asns* for CD8^+^ T cell activation, accompanied by T cell receptor (TCR) triggering and metabolic reprogramming via mTORC1 and Myc activation [49]. Dissecting the network perhaps enhances our understanding of the mechanisms for how the lncRNAs are involved in the peanut allergic response. 

Moreover, our samples merely came from peanut-induced allergic Brown rats in limited numbers. Further work is required to validate the diagnostic value of these lncRNAs in a large cohort of population samples. Despite continued advances and development of novel omics techniques, developing a definitive diagnostic test to negate the need for oral food challenges remains elusive. For clinical secondary use in the future, it should be pointed out that further investigation is necessary to eliminate the factors influencing pretest probability, such as age, population type, and residential environment.

## 5. Conclusions

The lncRNA-based signature is a novel tool that could provide simple and accurate clinical outcome prediction. In summary, our four-lncRNA signature could be a practical and reliable prognostic tool for peanut allergy, which can offer an incremental clinical value over the traditional system for diagnostic prediction, thus providing the underlying basis for observing and supervising desensitization therapy for peanut allergy. Additionally, the detailed regulatory mechanisms of lncRNAs involved in peanut allergy need to be elucidated in the future.

## Figures and Tables

**Figure 1 foods-11-03760-f001:**
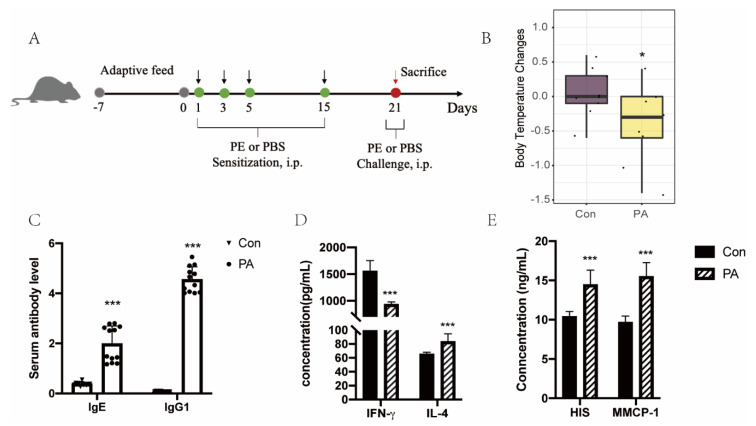
Peanut induced the Th2-type hypersensitivity response. (**A**) Schematic drawing for the Brown Norway rat systematical peanut allergy protocols applied in this study (N_Con_ = 18, N_PA_ = 23). (**B**) The body temperature changes after challenge on Day 21. (**C**) The secretion of peanut-specific antibody IgE and IgG1 between Con and PA groups. (**D**) The level of Th1 cytokine IFN-γ and Th2 cytokine IL-4. (**E**) Histamine and MMCP-1 release levels in PA individuals and their controls. Con: control group, PA: peanut allergy group. * *p* < 0.05, *** *p* < 0.001 (compared to the control group).

**Figure 2 foods-11-03760-f002:**
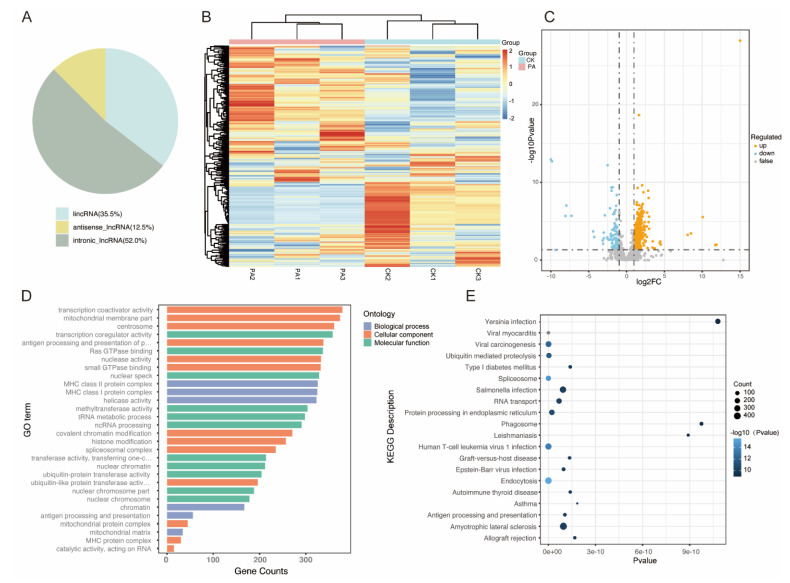
Identification of differentially expressed lncRNAs (DELs) between peanut allergy and control groups. (**A**) Classification and proportion of identified lncRNAs. (**B**) Hierarchical clustering for distinguishable lncRNA expression profiling among groups. (**C**) Volcano plot of all DELs between PA individuals (N = 3) and healthy controls (N = 3). (**D**,**E**) GO (**D**) and KEGG (**E**) enrichment pathway analysis involving DELs.

**Figure 3 foods-11-03760-f003:**
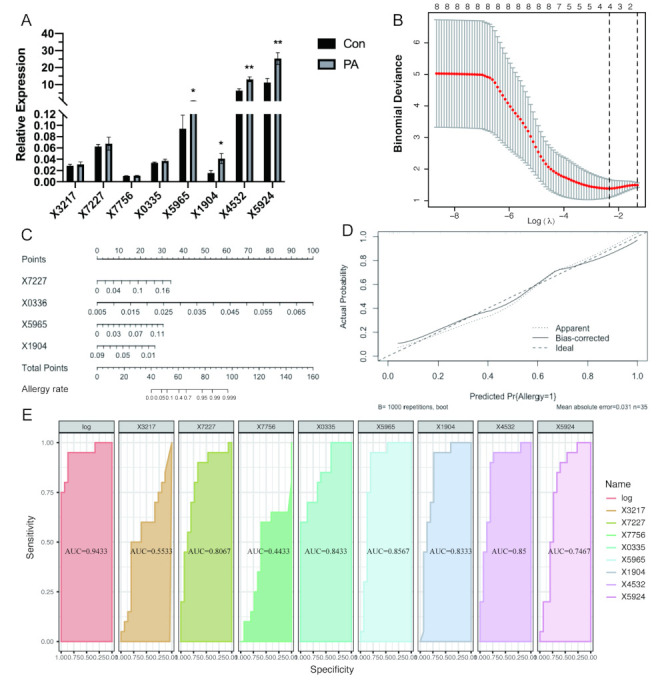
Construction of diagnostic model based on lncRNAs. (**A**) The expression levels of eight candidate lncRNAs verified by qRT-PCR (N_Con_ = 18, N_PA_ = 23). (**B**) LASSO analysis showed that four lncRNAs were associated with PA in the training set. (**C**) A nomogram to predict the probability of peanut allergy based on logistic regression. (**D**) Calibration curve for the constructed logistic regression equation. (**E**) Performance of individual candidate lncRNAs and four-lncRNA panel as biomarkers for PA exhibited by the receiver operating characteristic (ROC) curves in the training set (N = 3). * *p* < 0.05, ** *p* < 0.01 (compared to the control group).

**Figure 4 foods-11-03760-f004:**
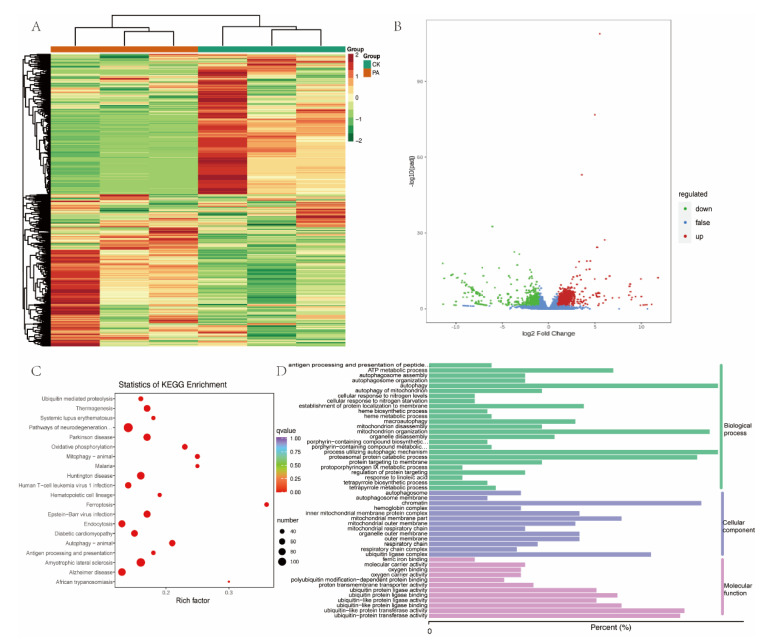
Differential expression of mRNAs between peanut allergy and control subjects. (**A**) heatmap showing profiles of mRNAs in the blood sample of peanut allergy group (N = 3) and control group (N = 3). (**B**) Volcano plot of all genes differentially expressed between two groups. (**C**) KEGG enrichment of differentially expressed genes. (**D**) The top 50 signaling pathways enriched by GO analysis with biological process, cellular components, and molecular function.

**Figure 5 foods-11-03760-f005:**
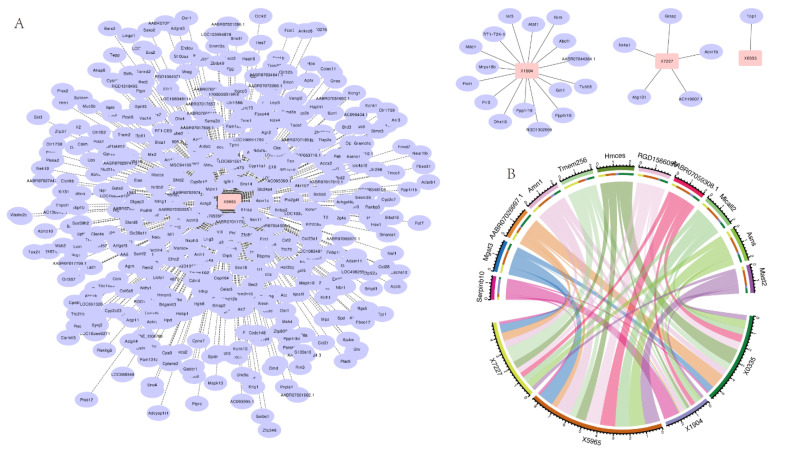
Correlation analysis between selected four lncRNAs and differentially expressed genes. (**A**) The predicted target genes (purple) of lncRNAs (pink) based on positional relationship (cis) and expression correlation (trans) with protein-coding genes. Solid lines represent cis-regulation and dashed lines represent trans-regulation. (**B**) Correlation between four lncRNAs and DEGs. The thickness of the bands show the degree of correlation between the genes.

**Table 1 foods-11-03760-t001:** The expressions of 8 candidate lncRNAs.

ID	GeneName	Log2FoldChange	*p*-Value	Regulated
ENSRNOT00000093217	*AABR07037436.1*	1.04783179	0.0192504	up
ENSRNOT00000087227	*AC119007.3*	1.3536378	0.03837092	up
ENSRNOT00000057756	*Cep128*	1.83746359	0.01298434	up
ENSRNOT00000090335	*AABR07054490.1*	1.22109388	0.0296379	up
ENSRNOT00000085965	*LOC100911851*	4.52387407	0.00970828	up
ENSRNOT00000081904	*Mrps18b*	1.41216762	0.00095665	up
ENSRNOT00000074532	*Snx3*	1.38905965	0.00420968	up
ENSRNOT00000075924	*Snx3*	1.64743931	2.17 × 10^−19^	up

**Table 2 foods-11-03760-t002:** Diagnostic model in logistic regression analysis.

ID	Coef	HR	95%CI	*p*-Value
ENSRNOT00000087227	38.5719	5.6437 × 10^16^	(0, 6.49 × 10^36^)	0.1017
ENSRNOT00000090335	313.8883	2.09 × 10^136^	(1191575465767.2, 3.66 × 10^260^)	0.0315
ENSRNOT00000085965	51.874	3.3775 × 10^220^	(0, 1.59 × 10^52^)	0.1367
ENSRNOT00000081904	−60.5057		(0, 69263846322464.2)	0.1992
Intercept	−8.9084			0.0123

**Table 3 foods-11-03760-t003:** Performance of four-lncRNA panel as biomarkers for peanut allergy.

Group	AUC	Specificity	Sensitivity	Threshold
PA vs. Con	0.75	1.0	0.5	−2.351
PA vs. MA	0.6667	1.0	0.667	−5.565
PA vs. SA	0.75	0.75	1.0	−6.37

## Data Availability

The data presented in the study are deposited in the NCBI Sequence Read Archive, accession number PRJNA841053.

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
