# Peer review of "Long Non-Coding RNAs Expressed in the Peanut Allergy for Understanding the Pathophysiology of Peanut Allergy Rat Model"

_foods, 2022, doi:10.3390/foods11233760_

Round 1
Reviewer 1 Report
The authors present a very interesting work on the search for biomarkers involved in peanut allergy. The work focuses on the development of an experimental model of peanut allergy in rats, the search and identification of lncRNAs involved in the phenomena of the allergic disease, and the elaboration of a predictive model for the diagnosis/prognosis of this allergy based on these biomarkers.
The study is very thorough using a methodology based on multi-omics and bioinformatics analysis. Although more extensive studies should be carried out, the results are promising and of future application in the prognosis and diagnosis of this food allergy.
However, there are some points that the authors could clarify/correct:
Introduction:
Line 34: “but there is no effective cure for food allergy”
The authors are right on this issue, but in recent years, great efforts are being made to alleviate this situation. Clinical trials of immunotherapy for peanut allergy, including Oral (OIT), epicutaneous (EPIT), and sublingual (SLIT) immunotherapy have been developed as potential treatments for peanut allergy and it seems that oral immunotherapy is one of the best approaches.
For example:
Kim EH, Patel C, Burks AW. Immunotherapy approaches for peanut allergy. Expert Rev Clin Immunol. 2020 Feb;16(2):167-174. doi: 10.1080/1744666X.2019.1708192. Epub 2020 Jan 12. PMID: 31928251.
Dunlop JH. Oral immunotherapy for treatment of peanut allergy. J Investig Med. 2020 Aug;68(6):1152-1155. doi: 10.1136/jim-2020-001422. Epub 2020 Jul 14. PMID: 32665367.
Mat & Met and Results:
Allergic Rat Models:
The authors established the peanut allergy model in a group of 23 rats belonging to the “training set” and used 18 individuals as a control group. However, the “validation set” was performed on 3 individuals. Did the authors use only 3 individuals to search for and identify lncRNAs? Please, explain this point
Results:
Line 166-172:
The authors describe the results of the experimental model setup and refer to Figure 2B-E. The reference should be to figure 1B-E, shouldn't it?
Line 181:
Is “lincRNA” the correct term, or should it be lncRNA? If so, please correct Figure 2A
Point 3.2. Construction and validation of the four-lncRNA signature:
The authors selected eight lncRNA candidates from 6342 total lncRNAs. The candidates were: X3217, X7227, X7756, X0335, X5965, X1904, X4532 and X5924 according to 3 criteria: up-regulated expression, known gene and the average relative expression validated by qRT-PCR > 0.01. Of 8 lncRNA it appears that 4 (X5965, X1904, X4532, X5924) were selected as the best biomarkers, but, at the end, X7227, X0335, X5965 and X1904 were selected because the better specificity. However, and if I understood correctly, X7227 and X0335 did not show statistically significant differences with the control group in terms of expression level. So why have they been selected as good biomarkers if they fail to meet one of the criteria to be candidates?
Minor:
Figure 1 C. Is there a unit for measuring antibody levels?
Figure 1 D. The first pair of columns is called IFN-r, it should not be IFN-É£?
Figure 5 A (left) does not show well
Line 294: “With Emerging….” "Emerging" should be in lowercase
Author Response
Thank you for your prompt reply and valuable suggestions. As mentioned in the answer below, we have further revised the article. Moreover, the revised part is highlighted in yellow in the resubmitted version.
Major:
- Thanks for your constructive suggestion. In recent years, immunotherapy for food allergy has indeed made great progress, which will provide a basis for the treatment of allergy. We have revised on line 34.
- In the training set, we established peanut allergy model in 23 rats, and used 18 individuals as control. Then we chose 3 rats to perform lncRNA sequencing and screen candidate lncRNAs in the control group and peanut allergy group, respectively. Next, we verified the candidate lncRNAs by qPCR and built a diagnostic model in 41 rats of training set. Finally, the verification was carried out in 12 rats in the validation set.
- Thank you for pointing out our mistakes in writing. We have corrected the error on Line 167.
- Some subtypes of lncRNAs have been identified: intergenic, intronic, antisense and overlapping lncRNAs. Among them, long intergenic non-coding RNAs (lincRNAs) represent the majority of lncRNAs and have been confirmed to play multiple biological roles. Figure 2A presented the classification of lncRNAs detected in rat blood samples, which is composed of 35.5% lincRNA, 12.5% antisense lncRNA and 52.0% intronic lncRNA.
Reference:
Ransohoff JD, Wei Y, Khavari PA. The functions and unique features of long intergenic non-coding RNA. Nat Rev Mol Cell Biol. 2018;19(3):143–57.
- Thank you for your careful reading of our manuscript. We validated the average relative expression of 8 candidate lncRNAs by qPCR. X7227 and X0335 in our model showed a certain degree of increase compared with the control group, although without significant differences, so we still regarded it as candidates. Most importantly, we focused on the diagnostic effect of the overall model.
Minor:
- The result in Figure 1C is the absorbance value measure at 450 nm, so there is not a concrete unit. And we are concerned about the difference between the control group and the peanut allergy group.
- It is our negligence that leads to writing mistakes, and we have corrected it in Figure 1D.
- We have replaced the new Figure 5 in the manuscript. In addition, the predicted target genes for X5965 were too many to display unclearly, so we will submit the list of relevant genes as supplementary materials.
- We are incredibly grateful for you to point the error out. We have corrected it as shown on line 291.
Reviewer 2 Report
Using a rat peanut allergy model, the authors analyzed blood lncRNAs during the induction of allergic symptoms by challenging peanut, and found a number of lncRNAs that increased or decreased in association with symptoms. Using these lncRNAs, the authors aimed to develop a more accurate diagnostic method for peanut allergy than is currently available in clinical practice, such as skin prick test, allergen-specific IgE test and double-blind challenge test.
Major point: There is a fundamental error in the design of this study. This is, the clinical diagnosis of food allergy is to accurately define the sensitization state to food allergens leading to the allergic symptoms, not to determine the condition of allergic symptoms. If the authors want to identify the lncRNAs associated with clinical diagnosis, they should analyze sensitized condition of rats allergic to peanut, not the rats that expressed allergic symptoms. Therefore, the significance of this study is to elucidate the pathophysiology involved in the development of peanut allergy symptoms, not to lead to a diagnosis. Therefore, The title of the manuscript should be “Long non-coding RNAs expressed in the peanut allergy for understanding the pathophysiology of rat peanut allergy model” and review the results and interpretation of the study.
Minor point 1: Abbreviations need to be corrected throughout the entire manual.
Minor point 2: There are several citation of old references, such as Ref. 3, 7, 8, 10.
Minor point 3: In Materials and Methods, 2.1. Allergic rat model section. 63 rats were used, but only 53 rats were used. What for 10 rats ?
Minor point 4: In Materials and Methods, 2.5. Bioinformatics section. References cited in this section should be described.
Author Response
Thank you for your time and valuable comments on the manuscript. All your valuable comments and suggestions helped us improve the article's quality. We are willing to do our best for the optimization of the manuscript. Moreover, the revised part is highlighted in green in the resubmitted version.
Major:
Thanks for your constructive advice. Our intention was to identify the lncRNAs associated with clinical diagnosis, but we confused the sensitized and elicited state of allergy. So, we have changed our title to “Long non-coding RNAs expressed in the peanut allergy for understanding the pathophysiology of peanut allergy rat model”.
Minor:
- Thanks for pointing out the non-standard format in our manuscript. We have made corrections in the text and abbreviations.
- Thanks very much for your valuable suggestion. We have replaced the old references with new research progress, such as Ref. 5, 10, 12.
- We are incredibly grateful for you to point this out. We have revised “63” to “53” on line 77.
- We have added relevant references to the bioinformatics section.
Round 2
Reviewer 2 Report
The revised manuscript was adequately corrected.